# Protective Effects of *Centella asiatica* Against Senescence and Apoptosis in Epidermal Cells

**DOI:** 10.3390/biology14020202

**Published:** 2025-02-14

**Authors:** Yu Tan, Ailing Hu, Jingya Lu, Yunhai Lin, Xuejing Li, Takuji Yamaguchi, Masahiro Tabuchi, Zenji Kawakami, Yasushi Ikarashi, Hiroyuki Kobayashi

**Affiliations:** Department of Personalized Kampo Medicine, Juntendo University Graduate School of Medicine, Tokyo 113-8421, Japan; ailing@juntendo.ac.jp (A.H.); j.lu.pv@juntendo.ac.jp (J.L.); y.lin.oa@juntendo.ac.jp (Y.L.); x.li.jw@juntendo.ac.jp (X.L.); tkyamagu@juntendo.ac.jp (T.Y.); m.tabuchi.oa@juntendo.ac.jp (M.T.); z.kawakami.sj@juntendo.ac.jp (Z.K.); y.ikarashi.zi@juntendo.ac.jp (Y.I.); koba@juntendo.ac.jp (H.K.)

**Keywords:** *Centella asiatica*, apoptosis, cellular senescence, oxidative stress, keratinocytes

## Abstract

The epidermis, the outermost layer of the skin, is primarily composed of keratinocytes. The dysfunction and senescence of keratinocytes, along with the destruction of the skin barrier due to apoptosis, can lead to a decline in various skin characteristics, such as regeneration, moisturization, elasticity, and stretchability, and promote skin aging, as evidenced by increased wrinkles and age spots, decreased moisturization, and reduced elasticity. Evidence suggests that 70% of these changes can be attributed to oxidative stress. *Centella asiatica*, a traditional medicinal plant, possesses potent antioxidant activity and may therefore help prevent cellular aging and exert antiapoptotic effects. However, no study has yet comprehensively elucidated these effects of *C. asiatica*. Our findings revealed that H_2_O_2_ causes cell death, oxidative stress, cellular senescence, and the apoptosis of human epidermal keratinocytes. Conversely, *C asiatica* promotes a significant decrease in all markers associated with H_2_O_2_-induced cell death, oxidative stress, cellular senescence, and apoptosis, highlighting its ability to prevent cellular senescence and apoptosis through its antioxidant activity. *C. asiatica*, with such a mechanism of action, holds considerable promise in the prevention and improvement of skin aging.

## 1. Introduction

Reactive oxygen species (ROS), such as singlet oxygen (^1^O_2_), superoxide anion radical (•O_2_^−^), hydrogen peroxide (H_2_O_2_), hydroxyl radical (•OH), and ozone (O_3_), are generated during respiration and by other factors, such as air pollution, ultraviolet light, smoking, strong mental stress, and strenuous exercise. Excessive ROS production induces cell damage; nonetheless, most cells have antioxidant defense mechanisms to protect against ROS damage, including endogenous antioxidant enzymes such as superoxide dismutase (SOD), glutathione peroxidase (GPx), and catalase (CAT), and exogenous antioxidants such as vitamin C or E, carotenoids, and catechins. Oxidative stress occurs when ROS production overwhelms the antioxidant defense capacity [1].

Excessive oxidative stress in keratinocytes [2,3], the main cell type in the epidermis, which is the skin’s outermost layer, causes irreversible DNA damage, dysfunction, cellular senescence, and ultimately apoptotic cell death [4]. The dysfunction and senescence of keratinocytes induce a decline in various skin characteristics, such as regeneration, moisturization, elasticity, and stretchability, and promote skin aging as evidenced by increased wrinkles and age spots, decreased moisturization, and decreased elasticity. Evidence suggests that 70% of these changes can be attributed to oxidative stress [3]. Therefore, preventing and ameliorating the series of oxidative stress-induced apoptosis in keratinocytes could certainly prevent and ameliorate skin aging.

*Centella asiatica* is a plant of the Apiaceae family whose leaf and stem extracts have been widely used as a traditional herbal medicine for skin diseases in China and Southeast Asian countries, with the following evidence reported [5,6,7]: *C. asiatica* extract attenuated periorbital wrinkles in female volunteers, ameliorated dermatitis in a mouse model of atopic dermatitis, increased the inductive property of hair by inhibiting three-dimensional spheroid cultured human dermal papilla cells, inhibited ultraviolet-induced oxidative stress and matrix metalloproteinase induction in keratinocytes, enhanced antioxidants in a rat skin wound model, promoted collagen production in dermal fibroblasts, facilitated skin whitening by suppressing ultraviolet-induced melanogenesis and inflammation in cocultured keratinocytes and melanocytes, and protected the skin by inhibiting glycation stress in melanocytes.

These findings suggest that these effects of *C. asiatica* closely involve antioxidant and associated anti-inflammatory effects. As mentioned previously, considering that 70% of skin aging is due to oxidative stress, *C. asiatica*, which has strong antioxidant capacity, may be able to prevent skin aging by preventing a series of oxidative stress-induced apoptoses in the skin. However, to the best of our knowledge, no study has yet examined in detail the effects of *C. asiatica* on the series of reactions involving oxidative stress-induced cellular senescence and apoptosis in keratinocytes. Scientifically demonstrating the series of cell death defense mechanisms of *C. asiatica* in skin epidermal keratinocytes will provide important evidence for the future use of *C. asiatica* extract as a preventive and improving agent for skin aging.

The current study therefore aimed to investigate the effects of *C. asiatica* extract on oxidative stress-induced cellular senescence and apoptosis in epidermal keratinocytes. For this purpose, we evaluated the action of *C. asiatica* by measuring cell viability as a marker of cell death; ROS, radical scavenging, SOD, GPx, and catalase activities as markers of oxidative stress; senescence-associated β-galactosidase (SA-β-gal) activity as a marker of cellular senescence; and caspase-3/9 activities and apoptotic cells as markers of apoptosis using an in vitro model of H_2_O_2_-induced oxidative stress in human keratinocytes (HaCaT cells).

## 2. Materials and Methods

### 2.1. C. asiatica Extract and H_2_O_2_

Dried crude drug pieces for *C. asiatica* decoction were purchased from Huanyan Medical Beauty Co., Ltd. (Anyue, China). *C. asiatica extract* was prepared at our laboratory (Department of Personalized Kampo Medicine, Juntendo University Graduate School of Medicine, Tokyo, Japan). Briefly, 30 g of dried herbal pieces were extracted with 300 mL of 50% ethanol at 50 °C for 8 h [8]. The extract was concentrated through rotary evaporation and adjusted to a final concentration of 1 mg/mL with dimethyl sulfoxide (DMSO: Sigma-Aldrich, St. Louis, MO, USA).

Approximately 124 chemical compounds including triterpenoids, polyphenolic compounds, and essential oils have been isolated and identified from *C. asiatica* [9]. Asiatic acid, madecassic acid, asiaticoside, and madecassoside are the major compounds identified in *C. asiatica* [10]. Our extract contained 0.12% asiatic acid, 0.54% madecassic acid, 0.25% asiaticoside, and 1.02% madecassoside [5].

Various concentrations (50–200 µg/mL) of *C. asiatica* extract used in the experiments were prepared using Dulbecco’s Modified Eagle Medium (DMEM: Gibco, Grand Island, NY, USA) containing 0.5% DMSO, which was used as the experimental medium. The concentrations of the extracts were set based on concentrations previously reported to have antioxidant effects in various cell culture experiments [7].

Previous reports have confirmed that 0.5% DMSO is a noncytotoxic concentration [11]. Our preliminary study also confirmed that the viability of HaCaT cells cultured in medium containing 0.5% DMSO was not significantly different from that of cells cultured in medium without DMSO.

H_2_O_2_, an oxidation inducer, was purchased from Katayama Chemical Industries Co., Ltd. (Osaka, Japan). The concentration of H₂O₂ used in this study, 300 μM, is widely acceptable as a concentration that induces oxidative stress-induced cell damage [12,13]. In this study, concentrations were set based on this information.

### 2.2. Materials for Cell Culture

The human keratinocyte cell line (HaCaT cells) was purchased from Cell Lines Service (DKFZ, Heidelberg, Germany). DMEM (CLS Cell Service GmbH, Eppelheim, Germany) supplemented with 10% fetal bovine serum (FBS) and 1% penicillin–streptomycin (PSMS) was then used as the culture medium (hereinafter called normal medium). FBS and PSMS used for the preparation were obtained from Gibco (Grand Island, NY, USA). DMEM containing 0.5% DMSO was used as the experimental medium for evaluating the efficacy of the drugs (hereinafter referred to as the experimental medium). DMSO was purchased from Sigma-Aldrich Co. LLC. (St. Louis, MO, USA). Trypsin used for cell passaging was purchased from Gibco.

Culture flasks (75 cm^2^) as well as collagen-coated 6- and 96-well microplates or 96-well black/clear-bottom microplates were purchased from Corning Life Sciences (Acton, MA, USA).

### 2.3. Assay Materials

Cell Counting Kit-8 (CCK-8), ROS Assay Kit, DPPH Antioxidant Assay Kit, SOD Assay Kit, Cellular Senescence Assay Kit, and Count Normalization Kit were purchased from Dojindo Laboratories (Kumamoto, Japan). Fluorometric Glutathione Peroxidase Assay Kit, Catalase Assay Kit, and Caspase-9 Activity Apoptosis Assay Kit were purchased from AAT Bioquest, Inc. (Pleasanton, CA, USA). Caspase-3 Activity Apoptosis Assay Kit was purchased from Cell Signaling Technology (Danvers, MA, USA). The 96-well microplate or black/clear-bottom microplates used for the assays were purchased from Corning Life Sciences. Alexa Fluor™ 488 Annexin V/Dead Cell Apoptosis Kit was purchased from Thermo Fisher Scientific (Waltham, MA, USA). Other chemicals were purchased from commercial sources.

### 2.4. Cell Culture

HaCaT cells (1 × 10^6^ cells) were seeded in normal medium (15 mL) in a 75 cm² collagen-coated flask and cultured in a CO_2_ incubator (MCO-170AIC-PJ, Panasonic, Osaka, Japan) at 37 °C and 5% CO_2_ until the cells reached full confluence. Thereafter, the cells were treated with 0.25% trypsin and passaged. Eighth-passage cells were then used in this study.

### 2.5. Experimental Design

The experimental procedure is shown in Appendix A. Eighth-passage HaCaT cells were seeded in normal medium in 75 cm^2^ flasks or different microplates depending on the experiment and cultured in a CO_2_ incubator (37 °C and 5% CO_2_). After the cells reached 80–90% confluence, they were divided into control, H_2_O_2_, *C. asiatica*, and *C. asiatica* + H_2_O_2_ groups for the initiation of the experiment (Day 0). On Day 0, the normal medium in the control group was replaced with the experimental medium and cultured for 2 days. In the *C. asiatica* group, the normal medium was replaced with the experimental medium containing *C. asiatica* extract (final concentrations: 50, 100, and 200 μg/mL) and cultured for 2 days. In the H_2_O_2_ group, the normal medium was replaced with the experimental medium and cultured for 1 day, followed by one more day in the experimental medium containing H_2_O_2_ (final concentrations: 75–600 μM for the cell viability experiment and 300 μM for the other experiments). In the *C. asiatica* + H_2_O_2_ group, the normal medium was replaced with the experimental medium containing *C. asiatica* extract (final concentrations: 50, 100, and 200 μg/mL) and cultured for 1 day, followed by one more day in the same medium containing *C. asiatica* extract (final concentrations: 50, 100, and 200 μg/mL) + H_2_O_2_ (final concentration: 300 μM). Cell viability, oxidative stress markers (ROS, radical scavenging, SOD, GP_X_, and CAT activities), cellular senescence markers (SA-β-gal activity), or apoptosis markers (caspase-9/3 activity and apoptotic cells) were measured on Day 2.

The cell viability experiments were evaluated in the control, H_2_O_2_, and *C. asiatica* groups using collagen-coated 96-well microplates (100 μL medium/well) seeded with 5 × 10^3^ cells/well. The other experiments were evaluated in the control, H_2_O_2_, and *C. asiatica* + H_2_O_2_ groups using different culture vessels and cell numbers depending on the assay. For the radical scavenging assay, 96-well microplates (100 μL medium/well) seeded with 5 × 10^3^ cells/well were used. For the ROS, SA-β-gal, and caspase-3 assays, collagen-coated 96-well black/clear-bottom microplates (100 μL medium/well) seeded with 5 × 10^3^ cells/well were used. For the apoptotic cell detection assay, collagen-coated 6-well microplates (2 mL medium/well) seeded with 3 × 10^4^ cells/well were used. For the SOD, catalase, GPx, and caspase-9 assays, collagen-coated 75 cm^2^ flasks (15 mL/flask) seeded with 1 × 10^6^ cells/flask were used.

For each experiment, two or three preliminary experiments were performed before the main experiment to confirm the reproducibility of the results.

### 2.6. Assay Methods

#### 2.6.1. Cell Death Marker

Cell viability as a cell death marker was measured using CCK-8 according to the assay kit manufacturer’s protocol [14]. Briefly, CCK-8 solution (10 μL/well) was added to culture wells and incubated for 2 h at 37 °C in a CO_2_ incubator, after which the absorbance at 450 nm was measured using a microplate reader (Thermo Fisher Scientific, MA, USA). Cell viability was calculated as the absorbance percentage of the cells treated with the test substances relative to that of the control.

#### 2.6.2. Oxidative Stress Markers

ROS, radical scavenging, SOD, GP_X_, and catalase activities were measured as oxidative stress markers.

##### ROS Activity

Intracellular ROS activity was measured using an ROS Assay Kit according to the manufacturer’s protocol [15]. Briefly, the medium in a 96-well black/clear-bottom culture plate was removed, after which the cells in the wells were washed twice with 100 µL of Hanks’ Balanced Salt Solution (HBSS). Thereafter, high-sensitivity DCFH-DA dye working solution (100 µL/well) was added, followed by incubation for 30 min in a CO_2_ incubator (37 °C and 5% CO_2_). After the working solution was removed, the cells were washed twice with HBSS, and fresh HBSS (100 µL/well) was added. ROS activity (measured using relative fluorescent units [RFU]) was assessed by measuring the fluorescence intensity at the excitation/emission wavelengths of 510/530 nm using a FlexStation^®^ multimode microplate reader (Molecular Devices, LLC, San Jose, CA, USA).

##### Radical Scavenging Activity

Radical scavenging activity in the medium in the 96-well culture plate was measured using a DPPH Antioxidant Assay Kit (Dojindo Laboratories) according to the manufacturer’s protocol [16]. Briefly, 100 µL of a working solution of DPPH (2,2-diphenyl-1-picrylhydrazyl), a stable radical, was added to 20 µL of sample medium and incubated at 25 °C in the dark for 30 min. The absorbance levels of the DPPH after incubation were measured at a wavelength of 517 nm using a microplate reader (Thermo Fisher Scientific). Radical scavenging activity (%) was expressed as the scavenging rate of DPPH during the 30 min incubation.

##### SOD Activity

After removing the medium in the 75 cm flask, the cells were washed twice with 5 mL of phosphate-buffered saline (PBS). The cells were then lysed via sonication (Branson Sonifire 250 advanced, Emerson Japan, Ltd., Tokyo, Japan) on ice for 3 min and centrifuged (10,000× *g*, 15 min), subsequently using the supernatant as the test sample. The SOD activity in the samples was measured using an SOD Assay Kit according to the manufacturer’s protocol [17]. Briefly, the test samples (20 µL) were transferred into a 96-well assay plate to which the WST-1 working solution (200 µL) and enzyme working solution (20 μL) were later added. After incubating the mixture at 37 °C for 20 min, the absorbance of the formed WST-1 formazan was measured at a wavelength of 450 nm using a Spectramax 340PC^384^ microplate reader (Molecular Devices). The SOD activity (U/mL) was calculated based on the 50% inhibition rate of WST-1 formazan produced by the enzyme working solution.

##### GPx Activity

The cell lysate supernatants obtained from the same procedure described in the SOD Activity section were used as test samples. GPx activity in the test samples was measured using a Fluorometric Glutathione Peroxidase Assay Kit according to the manufacturer’s protocol [18]. Briefly, 50 µL of the test samples or GPx standards were transferred to a 96-well black/clear-bottom assay plate, after which an equal volume of GPx working solution was added, followed by incubation at room temperature for 30 min. Thereafter, 20 µL each of NPDP probe and NADP assay solution were added and incubated at room temperature for 20 min. Finally, 15 µL of the enhancer solution was added and incubated at room temperature for 60 min, after which the fluorescence intensity was measured at the excitation/emission wavelengths of 420/480 nm using a FlexStation^®^ multimode microplate reader (Molecular Devices, LLC). GPx activity (mU/mL) was calculated from the calibration curve of GP_X_ standards.

##### Catalase Activity

The cell lysate supernatants obtained from the same procedure described in the SOD Activity section were used as test samples. The catalase activity in the test samples was measured using a Fluorometric Catalase Assay Kit [18]. Briefly, 50 µL of the test samples or catalase standards were transferred to a 96-well black/clear-bottom assay plate, after which an equal volume of H_2_O_2_ assay buffer was added and incubated at room temperature for 25 min. Thereafter, 50 µL of the catalase assay mixture was added and incubated at room temperature for 25 min, after which the fluorescence intensity was measured at the excitation/emission wavelengths of 540/590 nm using a FlexStation^®^ multimode microplate reader (Molecular Devices, LLC). Catalase activity (mU/mL) was calculated from the calibration curve of the catalase standards.

#### 2.6.3. Cellular Senescence Marker

SA-β-gal activity was measured as a cellular senescence marker [2]. In this assay, SA-β-gal activity ultimately needed to be normalized to the cell number. Therefore, cells in each of the wells in a 96-well black/clear-bottom culture plate were stained with Hoechst for nucleic acid using the Cell Count Normalization Kit according to the manufacturer’s protocol. Thereafter, the Hoechst-derived fluorescence intensity was measured at the excitation/emission wavelengths of 350/460 nm using a FlexStation^®^ multimode microplate reader (Molecular Devices, LLC). After measuring the Hoechst intensity, the cells were washed once with 100 μL of PBS, after which 50 μL of lysis buffer was added to each well and incubated at room temperature for 10 min. SA-β-gal activity in the lysates was measured using a Cellular Senescence Assay Kit according to the manufacturer’s protocol [19]. Briefly, 50 μL of SPiDER-βGal working solution was added to each well containing the lysate and incubated at 37 °C for 30 min. Subsequently, 100 μL of Stop Solution was added to each well, after which the SPiDER-βGal-derived fluorescence intensity was measured at the excitation/emission wavelengths of 535/580 nm using a FlexStation^®^ multimode microplate reader (Molecular Devices, LLC). SA-β-gal activity was expressed by normalization as the ratio of SPiDER-βGal/Hoechst intensity.

#### 2.6.4. Apoptosis Markers

Catalase-9/3 activities and apoptotic cells were measured as apoptotic markers.

##### Caspase-9 Activity

The cell lysate supernatants obtained from the same procedure described in the SOD Activity section were used as the test samples. Caspase-9 activity was measured using the Caspase-9 Activity Apoptosis Assay Kit [20]. Briefly, 100 µL of the test sample was transferred to a 96-well black/clear-bottom assay plate to which an equal volume of caspase-9 substrate working solution was added, followed by the incubation of the plate at room temperature for 1 h. Caspase-9 activity (RFU) was determined by measuring the fluorescence intensity using a FlexStation^®^ multimode microplate reader (Molecular Devices, LLC) at the excitation/emission wavelengths of 540/620 nm.

##### Caspase-3 Activity

Caspase-3 activity was measured using the Caspase-3 Activity Apoptosis Assay Kit [21]. Briefly, after removing the culture medium from a 96-well black/clear-bottom culture plate, the cells were washed with 100 μL of ice-cold PBS, and 30 μL of cell lysis buffer was added to the wells. The well-plate was placed on ice for 5 min, after which 200 µL of caspase-3 substrate solution was added to the wells. After incubation at 37 °C for 1 h in the dark, caspase-3 activity (RFU) was determined by measuring the fluorescence intensity at the excitation/emission wavelengths of 380/450 nm using a FlexStation^®^ multimode microplate reader (Molecular Devices, LLC).

##### Apoptotic Cells

Apoptosis was assessed using the Alexa Fluor™ 488 Annexin V/Dead Cell Apoptosis Kit, according to the manufacturer’s protocol [22]. Briefly, cells harvested from each well of the 6-well culture plate were transferred into a 15 mL test tube and washed twice with cold PBS (3 mL). After the cells were resuspended in 500 µL of binding buffer and the cell concentration was adjusted to approximately 3 × 10^4^ cells, 2.5 µL each of Alexa Fluor 488 Annexin V and propidium iodide (PI) working solutions were added, mixed gently, and placed on ice for 10 min. The stained cells were then analyzed using a flow cytometer (BD FACSCelesta™: BD Biosciences, San Jose, CA, USA). The cells, double-stained with Annexin V and PI, were finally presented as an Annexin V-PI scattergram divided into four quadrants (Q1–Q4): Q1 (normal live cells: Annexin V negative/PI negative, hereafter Annexin V^−^/PI^−^), Q2 (early apoptotic cells: Annexin V positive/PI negative, hereafter Annexin V^+^/PI^−^), Q3 (late apoptotic cells: Annexin V positive/PI positive, hereafter Annexin V^+^/PI^+^), and Q4 (necrotic cells: Annexin V negative/PI positive, hereafter Annexin V^−^/PI^+^). The percentage of the cell population in each quadrant relative to the total number of cells detected in the flow cytometry analysis was calculated using BD and FlowJo ver.10.8.1analysis software (BD Biosciences).

### 2.7. Statistical Analysis

All data were presented as means ± standard errors of the mean (SEM). After confirming the normal distribution of the data, the significance of associations was determined by one-way analysis of variance and post hoc analysis with Tukey’s multiple comparison test. Statistical significance was set at *p* < 0.05. These statistical analyses were performed using GraphPad Prism 9 (San Diego, CA, USA).

## 3. Results

### 3.1. Cell Death Prevention Effects of C. asiatica Extract

The effects of H_2_O_2_, *C. asiatica* extract, and H_2_O_2_ + *C. asiatica* extract on the viability of HaCaT cells are shown in Figure 1. Compared to controls, H_2_O_2_ (75–600 µM: 84–20.4%, *p* < 0.01–0.001) significantly decreased cell viability in a dose-dependent manner (Figure 1a), indicating that it induces cell death. Although *C. asiatica* (50–200 µg/mL) did not significantly affect cell viability (Figure 1b), it (50–200 µg/mL: 68.7–89.8%, *p* < 0.001) did significantly suppress the decrease in cell viability (i.e., cell death) induced by 300 µM of H_2_O_2_ (31.8%, *p* < 0.001) (Figure 1c).

### 3.2. Antioxidative Stress Effects of C. asiatica Extract

Figure 2 presents the effects of *C. asiatica* extract (50–200 µg/mL) on 300 µM H_2_O_2_-induced changes in the activities of the oxidative stress markers: ROS, RA, SOD, GP_X_, and catalase. Compared to controls (37.3 RFU), H_2_O_2_ significantly increased ROS activity (428.4 RFU, *p* < 0.001), which was significantly suppressed by cotreatment with *C. asiatica* (50–200 µg/mL: 381.7–293.9 RFU, *p* < 0.05–0.01) in a dose-dependent manner (Figure 2a). In contrast, H_2_O_2_ significantly decreased the radical scavenging activity (27.3%, *p* < 0.05) compared to controls (34.8%), whereas *C. asiatica* (50–200 µg/mL: 28.0–42.6%, *p* < 0.01 at 200 µg/mL) significantly ameliorated this decrease in scavenging activity (Figure 2b). Similarly, although H_2_O_2_ significantly decreased SOD (6.1 U/mL, *p* < 0.001 vs. controls 8.6 U/mL), GPx (4.1 mU/mL, *p* < 0.001 vs. controls 4.9 mU/mL), and catalase activities (51.6 mU/mL, *p* < 0.001 vs. 60.3 mU/mL), cotreatment with *C. asiatica* (50–200 µg/mL: SOD 8.1–9.1 U/mL, GPx 4.8–5.2 mU/mL, and catalase 58.2–60.8 mU/mL) ameliorated the decreases in the mentioned enzyme activities (*p* < 0.001; Figure 2c–e).

### 3.3. Anti-Cellular Senescence Effect of C. asiatica Extract

Figure 3 details the effects of *C. asiatica* extracts on the 300 µM H_2_O_2_-induced increase in SA-β-gal activity, a cellular senescence marker. Compared to controls (ratio: 0.327), H_2_O_2_ significantly increased SA-β-gal activity (ratio: 0.454, *p* < 0.001), which was significantly suppressed by cotreatment with *C. asiatica* (ratio: 50–200 µg/mL: 0.350–0.288, *p* < 0.01–0.001) in a dose-dependent manner.

### 3.4. Antiapoptotic Effects of C. asiatica Extract

Figure 4 presents the effects of *C. asiatica* extract on the H_2_O_2_-induced increase in apoptosis initiator caspase-9 and executioner caspase-3 activities. Compared to controls, 300 µM H_2_O_2_ significantly increased both caspase-9 activity (696.3 RFU vs. control 539.3 RFU, *p* < 0.01) and caspase-3 activity (599.4 RFU vs. control 452.4 RFU, *p* < 0.001). However, *C. asiatica* (50–200 µg/mL) significantly ameliorated the increased activities of caspase-9 (579.9–525.6 RFU, *p* < 0.05–0.01) and caspase-3 (478.4–468.3 RFU, *p* < 0.001) (Figure 4a,b).

Figure 5 depicts the typical flow cytometry quadrant scattergrams of the control, 300 µM H_2_O_2_, and 50–200 µg/mL *C. asiatica* + 300 µM H_2_O_2_ groups (Figure 5a–e), including quantitative data comparing the percentages of normal cell (Q1: Annexin V^−^/PI^−^), early apoptotic cell (Q2: Annexin V⁺/PI^−^), and late apoptotic cell (Q3: Annexin V⁺/PI⁺) populations among the groups (Figure 5f–h). Compared to controls (89.0%), 300 µM H_2_O_2_ significantly decreased the percentage of normal cells (71.6%, *p* < 0.01, *p* < 0.01), but *C. asiatica* (50–200 µg/mL: 88.9–93.2%, *p* < 0.01) significantly ameliorated such a decrease. In contrast, H_2_O_2_ significantly increased the percentages of early (5.21% vs. controls 2.36%, *p* < 0.001) and late apoptotic cells (24.03% vs. controls 8.23%, *p* < 0.001) compared to controls, whereas *C. asiatica* (50–200 µg/mL: early apoptotic cells 3.24–2.16%, *p* < 0.01–0.001 and late apoptotic cells 4.54–3.48%, *p* < 0.001) significantly ameliorated the increased percentages of both stages of apoptotic cells. The number of necrotic cells (Annexin V^−^/PI^+^) in the Q4 fraction of each group accounted for 0.6% to 1.1% of the total cell number, but no significant difference was observed between the groups .

## 4. Discussion

The current study has been the first to demonstrate that *C. asiatica* possesses a protective effect against oxidative stress-induced cell damage, such as cellular senescence and apoptotic cell death, in keratinocytes, which constitute over 90% of the outer layer of the skin, called the epidermis.

We initially examined the effects of the oxidative stress inducer H_2_O_2_ and the test substance *C. asiatica* on the viability of HaCaT cells (Figure 1). Notably, our findings showed that H_2_O_2_ (75–600 µM) dose-dependently reduced cell viability, indicating that H_2_O_2_ induces cell death at this particular range of concentration. Based on this result, we selected 300 µM as the optimal concentration to reliably damage HaCaT cells. In contrast, *C. asiatica* (50–200 µg/mL) demonstrated no significant effect on cell viability; however, when combined with 300 µM H_2_O_2_, it dose-dependently suppressed H_2_O_2_-induced cell death. This result suggests that the protective effect of *C. asiatica* against cell death specifically involves the H_2_O_2_-induced cell death response. Therefore, to elucidate the underlying mechanism of the cytoprotective effects of *C. asiatica*, detailed experiments were conducted using a 300 µM H_2_O_2_-induced cell death model.

Although oxidative stress induced by excessive ROS generation causes cell damage, many cells have antioxidant defense mechanisms that protect cells from ROS damage by activating antioxidant enzymes such as SOD, GPx, and catalase [1]. Therefore, to clarify the mechanism underlying H_2_O_2_-induced cell death and the protective effect of *C. asiatica* illustrated in Figure 1, it is necessary to clarify the balance between ROS production and defense, i.e., the oxidative stress state. In this experiment, to examine the involvement of oxidative stress, we used the balance between ROS and radical scavenging activities as well as the activity of antioxidant enzymes such as SOD, GPx, and catalase as indicators. As a result, it was found that under H_2_O_2_ exposure conditions wherein cell death was observed, intracellular ROS activity increased and radical scavenging activity decreased (Figure 2a,b). This result indicates that H_2_O_2_ clearly induced oxidative stress in the cells, which is strongly supported by the decrease in the activities of the antioxidant enzymes SOD, GPx, and catalase. In contrast, *C. asiatica* ameliorated the changes in all markers reflecting H_2_O_2_-induced oxidative stress (Figure 2c–e), demonstrating the protective effect of *C. asiatica* against oxidative stress. These results reveal that the antioxidant effects of *C. asiatica* [7,23] also apply to HaCaT keratinocytes.

In normal cells, oxidative stress damages the DNA. To prevent the further proliferation of abnormal cells that cannot repair the damaged DNA, the cells irreversibly stop dividing, triggering cellular senescence. Given that SA-β-gal is overexpressed in such senescent cells, it has been widely used as a marker of senescence [2,24]. In this study, SA-β-gal activity was significantly increased under H_2_O_2_-induced oxidative stress conditions, with *C. asiatica* significantly improving such an increase (Figure 3). This result suggests that *C. asiatica* suppressed oxidative stress-induced cellular senescence.

Dysfunctional senescent cells that suffer irreversible DNA and mitochondrial damage due to oxidative stress are eventually eliminated by apoptosis [25]. In particular, oxidative stress and DNA damage have been shown to activate the p53 protein, thereby impairing the permeability of the outer mitochondrial membrane and releasing cytochrome c into the cytoplasm. Cytochrome c then activates the apoptosis initiator caspase-9 and the executioner caspase-3, inducing apoptosis [11]. Accordingly, our findings showed that H_2_O_2_ increased the activities of both intracellular caspase-9 and -3 and that *C. asiatica* significantly inhibited the increase in such activities (Figure 4), suggesting that H_2_O_2_-induced cell death is apoptotic cell death and that *C. asiatica* suppressed apoptosis.

Apoptosis can also be identified through structural changes in the cell membrane detected using a flow cytometer with fluorescently labeled Annexin V and PI, which binds to DNA [26]. Normal cells without apoptosis contain phosphatidylserine (PS) on the inner side of the cell membrane. In early apoptotic cells, PS is exposed on the cell membrane surface. Early apoptotic cells are detected by binding Annexin V to this exposed PS. At this stage, the cell membrane has yet to become damaged; therefore, PI, which binds to DNA, is not taken up into the cell. However, in late apoptotic cells, whose apoptosis has progressed, the structure of the cell membrane is broken, PI flows into the cell, and the nucleus is stained. As such, both Annexin V and PI were used to stain late apoptotic cells. As shown in Figure 5, H_2_O_2_ significantly increased the percentage of early and late apoptotic cells, whereas *C. asiatica* significantly inhibited the increase in apoptotic cells. Together with previous results on caspase activity, our results certainly support the notion that H_2_O_2_ exposure induces apoptotic cell death and that the cell-death-inhibitory effects of *C. asiatica* can be attributed to its antiapoptotic effect.

As demonstrated in the current study, the series of results from H_2_O_2_ exposure indicate that oxidative stress caused by H_2_O_2_ triggers cell senescence and apoptotic cell death, which is in good agreement with previously reported results [11]. This highlights the appropriateness of the experimental conditions used to evaluate the effects of the test substance *C. asiatica* on H_2_O_2_-induced cell damage, as well as the high reliability of the effects of *C. asiatica* obtained under those conditions.

Taken together, the above-demonstrated results clearly indicate that the mechanism underlying the protective effect of *C. asiatica* extract against H_2_O_2_-induced cell death involves a series of mechanisms that protect cells from oxidative stress, thereby preventing cellular senescence and apoptosis.

A review of the safety and side effects of *C. asiatica* extract [5] showed that standardized extracts of 250 mg and 500 mg were well tolerated in single and multiple oral doses in clinical trials. Acute and subchronic oral toxicity studies in rats as well as microbial mutagenicity tests (Ames tests) have confirmed that *C. asiatica* extract is safe. However, clinical cases of jaundice, allergic reactions, and burning sensations have also been reported after taking *C. asiatica*. Thus, although the safety of *C. asiatica* has been demonstrated in preclinical studies, more clinical trials are warranted, given the side effects reported in clinical cases, albeit in a small number.

This study had some limitations. First, the detailed molecular mechanism of action of *C. asiatica* could not be clarified. Evidently, the antioxidant activity of *C. asiatica* is due to the activation of enzymes such as SOD, GPx, and catalase measured in this study. However, the involvement of glutathione [10], peroxiredoxin family [27], and signal transduction pathways [12] such as Nrf2/ARE and MAPK, both of which are closely related to antioxidant activity, cannot be ruled out. Future research should investigate whether the antioxidant effects of *C. asiatica* are mediated by these pathways to clarify its mechanism of action. We also demonstrated that *C. asiatica* suppresses oxidative stress-induced caspase-9/3 activation and apoptotic cell death. However, apoptosis is triggered by a series of upper regulatory pathways, including p53 protein expression, Bax activation and Bcl-2 inactivation, mitochondrial outer membrane permeability disorder, and cytochrome c release [4]. Therefore, the effects of *C. asiatica* on these trigger reactions and their relationships with the apoptosis suppression effect remain unclear.

Second, this study focused on the effects of CA extract. Unfortunately, we could not yet verify the constituent crude drugs and ingredients in this study. Plants and herbal medicines contain several bioactive components, such as phenols, flavonoids, polyphenols, and triterpenes, which have antioxidant and anti-inflammatory properties [2,28]. In the case of *C. asiatica*, approximately 124 compounds have been isolated and identified, including triterpenoids, polyphenolic compounds, and essential oils [9]. In particular, *C. asiatica* has been reported to contain triterpenes, such as asiaticoside, asiatic acid, madecassoside, and madecassic acid, which have strong antioxidant properties [8,9]. In humans, the triterpenoid glycosides asiaticoside and madecassoside are converted into their aglycons, asiatic acid and madecassic acid, through the action of the intestinal bacteria β-glucosidase, absorbed into the blood and distributed to many organs, including the skin [5]. Therefore, we speculate that the aglycone may be one of the potential active components responsible for the anti-cellular-senescence and antiapoptotic properties of *C. asiatica*. While the current study focused on the effects of *C. asiatica*, several active components, including triterpenes, should be investigated using the analytical methods used in this study in the future.

Finally, it is necessary to conduct in vivo experiments to verify whether the in vitro results obtained in this study are reflected in the prevention or improvement of skin aging. Older people [29] and aged mice [30] show signs of skin aging, including decreased skin elasticity, loss of barrier function, dry skin, and reduced skin sensation. Therefore, future in vivo experiments should be conducted in humans and animals.

The antioxidant effects of *C. asiatica* can help treat various diseases related to oxidation, such as digestive disorders, wounds, nervous system disorders, and circulatory disorders [5,6,7]. Although the effects of *C. asiatica* on wrinkles around the eyes, atopic dermatitis, hair, collagen, and skin whitening have been reported, more scientific evidence regarding its effects on the skin are needed. This is the first study to investigate the effects of *C. asiatica* on keratinocyte aging, which accounts for >90% of the epidermis, and demonstrate that *C. asiatica* can prevent cellular aging and act as an antiapoptotic agent, counteracting some of the causes of skin aging. Despite the above limitations, the results of this study have potential clinical and industrial applications aiming at preventing skin aging.

## 5. Conclusions

*C. asiatica* significantly decreased all markers of H_2_O_2_-induced cell death, oxidative stress, cellular senescence, and apoptosis in human epidermal keratinocyte HaCaT cells. This study has been the first to demonstrate that *C. asiatica* prevents oxidative stress-induced cellular senescence and apoptosis through its antioxidant activity. With such a mechanism of action, *C. asiatica* holds considerable promise in the prevention and improvement of skin aging.

## Figures and Tables

**Figure 1 biology-14-00202-f001:**
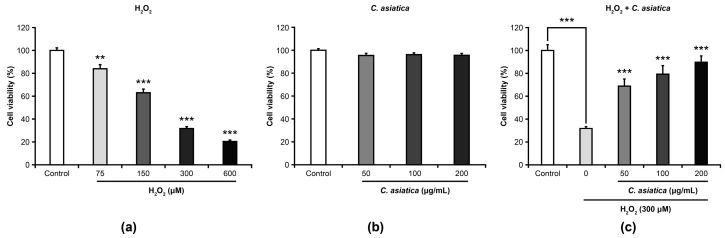
Effects of H_2_O_2_ (**a**), *C. asiatica* extract (**b**), and H_2_O_2_ + *C. asiatica* extract (**c**) on cell viability. Each value represents mean ± the standard error of mean (SEM) (*n* = 6). ** *p* < 0.01, and *** *p* < 0.001 vs. 300 µM H_2_O_2_ (one-way analysis of variance + Tukey’s multiple test).

**Figure 2 biology-14-00202-f002:**
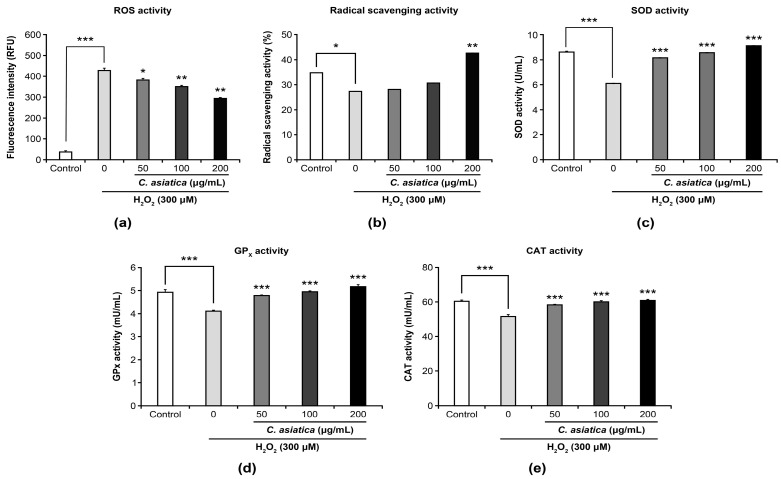
Ameliorative effects of *C. asiatica* extract on H_2_O_2_-induced changes in oxidative stress marker activities: (**a**) reactive oxygen species (ROS), (**b**) radical scavenging, (**c**) superoxide dismutase (SOD), (**d**) glutathione peroxidase (GPX), and (**e**) catalase activity. Each value represents mean ± SEM (*n* = 3). * *p* < 0.05, ** *p* < 0.01, and *** *p* < 0.001 vs. 300 µM-H_2_O_2_ (one-way analysis of variance + Tukey’s multiple test).

**Figure 3 biology-14-00202-f003:**
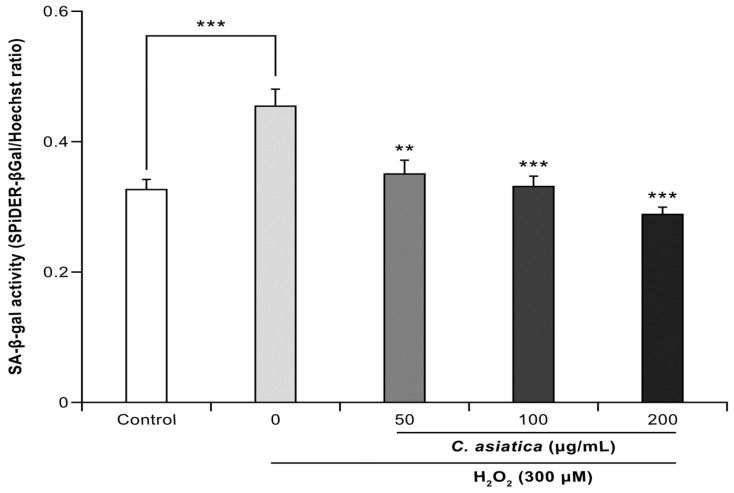
The ameliorative effect of *C. asiatica* extract on H_2_O_2_-induced increase in SA-β-gal activity, a marker of cellular senescence. Each value represents the mean ± the SEM (*n* = 6). ** *p* < 0.01, and *** *p* < 0.001 vs. 300 µM H_2_O_2_ (one-way analysis of variance + Tukey’s multiple test).

**Figure 4 biology-14-00202-f004:**
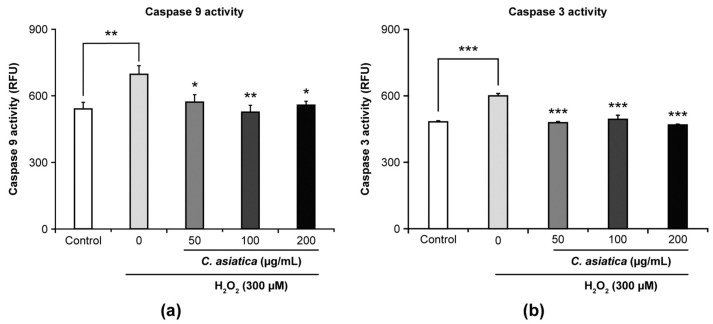
Ameliorative effects of *C. asiatica* extract on H_2_O_2_-induced increase in apoptotic marker activity: (**a**) caspase-9 and (**b**) caspase-3. Each value represents mean ± SEM (*n* = 6). * *p* < 0.05, ** *p* < 0.01, and *** *p* < 0.001 vs. 300 µM H_2_O_2_ (one-way analysis of variance + Tukey’s multiple test).

**Figure 5 biology-14-00202-f005:**
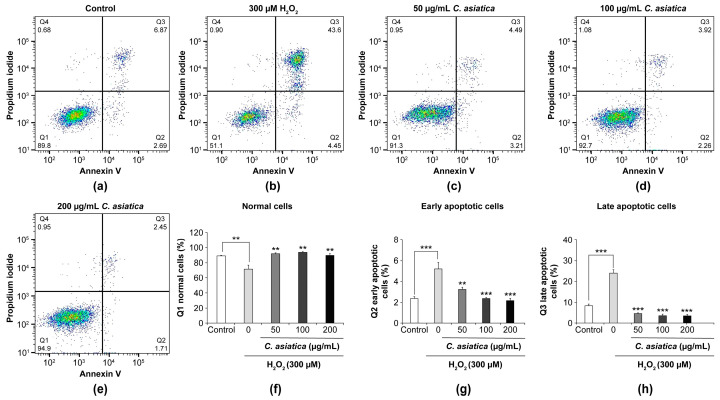
The ameliorative effects of *C. asiatica* extract on H_2_O_2_-induced apoptotic cells. Panels (**a**–**e**) show the flow cytometry of the four-quadrant Annexin V–PI scattergrams in controls, H_2_O_2_ (300 µM)-treated, and *C. asiatica* (50–200 µg/mL) + H_2_O_2_ combination-treated cells. Quadrants Q1, Q2, Q3, and Q4 represent normal (Annexin V^−^/PI^−^), early apoptotic (Annexin V⁺/PI^−^), late apoptotic (Annexin V⁺/PI⁺), and necrotic (Annexin V^−^/PI⁺) cell populations. Panels (**f**–**h**) show the quantitative data comparing the percentages of normal, early, and late apoptotic cell populations among the groups. Each value represents the mean ± the SEM (*n* = 3). ** *p* < 0.01, and *** *p* < 0.001 vs. 300 µM H_2_O_2_ (one-way analysis of variance + Tukey’s multiple test).

## Data Availability

The data presented in this study are available from the corresponding author upon reasonable request.

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
