# Peer review of "Protective Effects of Centella asiatica Against Senescence and Apoptosis in Epidermal Cells"

_biology, 2025, doi:10.3390/biology14020202_

Round 1

Reviewer 1 Report

Comments and Suggestions for Authors

The research work entitled "Protective effects of Centella asiatica extract against oxidative stress-induced cellular senescence and apoptosis through antioxidant activity in human epidermal keratinocytes" seemed to me to be a well-structured work and explores a relevant topic in the field of biology, physiology and medicine since it addresses various topics of interest.

From my point of view, this work "seems to have been carried out under methodological rigor" with solid results but with some details that I will question point by point and that may be areas for improvement to strengthen the impact of the work and favor the reading of the manuscript.

Title: This is clear and descriptive, but could benefit more from being simplified to better engage the reader, as a long title can be tiring from the start of the reading.

Abstract: This provides an overview of the study, but it would be useful to include specific values ​​of the findings to reinforce the accuracy of the results already in the abstract and make for a more fluid and engaging read.

The introduction clearly explains the study problem and its relevance and also adequately justifies the choice of the Centella asiatica species as the object of study and clearly relates the effects of ROS.

Some recommendations for the introduction would be to incorporate more recent references (post-2020) to strengthen the context, reduce redundancies when describing the role of ROS in cellular aging, identify how this research complements and/or fills gaps in previous research, and describe how this study can help further studies "a future perspective".

In the materials and methods section, the procedures are described in detail and this allows their replication. The choice of markers of oxidative stress and apoptosis is appropriate.

Provide a more detailed justification for the selected concentrations of H2O2 and Centella asiatica. Clarify whether additional controls were performed to rule out possible interference from DMSO. Specify whether measures were taken to minimize experimental variations (e.g., replication in different cell lines).

Another thing that really catches my attention is the "catalase activity" since there are other enzymes that can consume or reduce H2O2, as in this example CATALASE, we also have the PEROXYREDOXIN family, so I strongly recommend listing how to consume H2O2.

I sincerely think it is inappropriate to include a figure in the methodology. I recommend sending this Figure 1 to supplementary materials, since it does not provide a contribution to the research.

In the RESULTS section, they are correctly presented in tables and figures. The statistical analyses are appropriate for this study and are in a clear context. Only for reinforcement, the exact values ​​should be included in the text to complement the graph.

They could explain how these results can have implications at clinical and/or industrial level.

DISCUSSION: The relevance of the results is analyzed in a detailed and adequate manner, establishing clear connections with previous studies. However, there are some points that I would like you to consider:

1. Only that the possible molecular mechanisms underlying the observed effects of Centella asiatica are speculated "EXPLAIN BETTER AND HAVE A CLEAR LINE".

2. Propose specific future experiments to validate the hypotheses raised.

3. Discuss limitations of the study, such as the in vitro approach, and suggest in vivo studies to corroborate the findings.

4. Expand the discussion on the biological implications of changes in antioxidant markers.

In the ETHICS section, it is appropriately stated that there was no external funding or conflicts of interest. However, it would be ideal to mention whether ethical approval was obtained for the use of cell lines; if YES, the Ethics Committee and the number of the approved protocol for said work should be included.

Finally, review grammar and style to ensure fluency and accuracy in the English language.

Comments on the Quality of English Language

Just improve the way of writing so that it has better fluency when reading it.

Author Response

Dear Reviewer 1:

We would like to extend our sincere thanks for your thorough review and positive feedback on our manuscript. Based on your comments, we have carefully revised the manuscript. All revisions are shown in red in the revised manuscript.

Below is a list of the revisions we have made based on your feedback.

POINT-BY-POINT RESPONSES TO THE REVIEWER 1 COMMENTS

Comment 1

Title: This is clear and descriptive, but could benefit more from being simplified to better engage the reader, as a long title can be tiring from the start of the reading.

Response to Comment 1

We agree with your comment. We have revised the title of this study to “Protective effects of Centella asiatica against senescence and apoptosis in epidermal cells (lines 2–3).

Comment 2

Abstract: This provides an overview of the study, but it would be useful to include specific values of the findings to reinforce the accuracy of the results already in the abstract and make for a more fluid and engaging read.

Response to Comment 2

Thank you for your valuable opinion. However, due to the character limit of the Abstract section, it is difficult to include the specific numerical values of the survey results in the abstract. However, in accordance with your comment 5, we have included the exact values of the measurement data in the text (lines 326–404 in the revised manuscript).

Comment 3

The introduction clearly explains the study problem and its relevance and also adequately justifies the choice of the Centella asiatica species as the object of study and clearly relates the effects of ROS.

Some recommendations for the introduction would be to incorporate more recent references (post-2020) to strengthen the context, reduce redundancies when describing the role of ROS in cellular aging, identify how this research complements and/or fills gaps in previous research, and describe how this study can help further studies "a future perspective".

Response to Comment 3

Thank you for your valuable feedback. Throughout the text, we have removed or updated references older than 2020 to newer ones where possible (see the References section of the revised manuscript).

Following the reviewers’ comments, we have re-edited the introduction to reduce redundancy in the explanations of cellular senescence and ROS and to clarify the novelty and future prospects of this research (lines 44–90 of the revised manuscript).

Comment 4

In the materials and methods section, the procedures are described in detail and this allows their replication. The choice of markers of oxidative stress and apoptosis is appropriate.

  • Provide a more detailed justification for the selected concentrations of H2O2 and Centella asiatica.
  • Clarify whether additional controls were performed to rule out possible interference from DMSO.
  • Specify whether measures were taken to minimize experimental variations (e.g., replication in different cell lines).
  • Another thing that really catches my attention is the "catalase activity" since there are other enzymes that can consume or reduce H2O2, as in this example CATALASE, we also have the PEROXYREDOXIN family, so I strongly recommend listing how to consume H2O2.
  • I sincerely think it is inappropriate to include a figure in the methodology. I recommend sending this Figure 1 to supplementary materials, since it does not provide a contribution to the research.

Response to Comment 4

  • On the selected concentrations of H2O2 and Centella asiatica (Please see lines 100–117 in the revised manuscript): The concentration of Hâ‚‚Oâ‚‚ used in this study, 300 μM, is widely acceptable as a concentration that induces oxidative stress-induced cell damage (12, 13). Based on this information, in the present study, we investigated the effects of 75–600 mM H2O2 on HaCaT cells.

Various concentrations (50–200 µg/mL) of C. asiatica extract used in the experiments were prepared using Dulbecco’s Modified Eagle Medium (DMEM) containing 0.5% DMSO. The concentrations of the extracts were set based on concentrations previously reported to have antioxidant effects in various cell culture experiments [7].

  • On the possible interference by DMSO (lines 110–113): Previous reports have confirmed that 0.5% DMSO is a noncytotoxic concentration [11]. Our preliminary study also confirmed that the viability of HaCaT cells cultured in medium containing 0.5% DMSO was not significantly different from that of cells cultured in medium without DMSO.
  • On experimental variations (lines 182–183): For each experiment, two to three pilot experiments were performed before the main experiment to check the variability (or variations) and reproducibility of the results.
  • On catalase activity and peroxiredoxin family: Thank you for your valuable information. In this study, we measured catalase and glutathione peroxidase as hydrogen peroxide-removing enzymes. As you pointed out, the peroxiredoxin family is also important as a hydrogen peroxide-removing enzyme (28). This information will be useful for future research. Therefore, in the Discussion section of the revised manuscript, we have mentioned the peroxiredoxin family as a future study (lines 430–445).
  • We agree. Figure 1 in the previous version has been changed to Supplement 1 in the revised manuscript (Please see Supplement 1).

Comment 5

  • In the RESULTS section, they are correctly presented in tables and figures. The statistical analyses are appropriate for this study and are in a clear context. Only for reinforcement, the exact values should be included in the text to complement the graph.
  • They could explain how these results can have implications at clinical and/or industrial level.

Response to Comment 5

  • Following the comments, we have included the exact values of the measurement data in the text (lines 326–404 in the revised manuscript).
  • Thank you for your valuable feedback. We addressed the potential clinical and industrial implications of our findings in the Discussion section of the revised manuscript as follows (lines 523–538):

The antioxidant effects of C. asiatica can help treat various diseases related to oxidation, such as digestive disorders, wounds, nervous system disorders, and circulatory disorders [5-7]. Although the effects of C. asiatica on wrinkles around the eyes, atopic dermatitis, hair, collagen, and skin whitening have been reported, more scientific evidence regarding its effects on the skin are needed. This is the first study to investigate the effects of C. asiatica on keratinocyte aging, which account for >90% of the epidermis, and demonstrate that C. asiatica can prevent cellular aging and act as an antiapoptotic agent, counteracting some of the causes of skin aging. Despite the above limitations, the results of this study have potential clinical and industrial applications aiming at preventing skin aging.

Comment 6

DISCUSSION: The relevance of the results is analyzed in a detailed and adequate manner, establishing clear connections with previous studies. However, there are some points that I would like you to consider:

  • Only that the possible molecular mechanisms underlying the observed effects of Centella asiatica are speculated "EXPLAIN BETTER AND HAVE A CLEAR LINE".
  • Propose specific future experiments to validate the hypotheses raised. Discuss limitations of the study, such as the in vitro approach, and suggest in vivo studies to corroborate the findings. Discuss limitations of the study, such as the in vitro approach, and suggest in vivo studies to corroborate the findings.
  • Expand the discussion on the biological implications of changes in antioxidant markers.
  • In the ETHICS section, it is appropriately stated that there was no external funding or conflicts of interest. However, it would be ideal to mention whether ethical approval was obtained for the use of cell lines; if YES, the Ethics Committee and the number of the approved protocol for said work should be included.
  • Finally, review grammar and style to ensure fluency and accuracy in the English language.

Response to Comment 6

  • We concluded the mechanism underlying the cell death protective effect of asiatica, at least, from the results of this study (lines 482–485):

Taken together, the above demonstrated results clearly indicate that the mechanism underlying the protective effect of C asiatica against H2O2-induced cell death involves a series of mechanisms that protect cells from oxidative stress, thereby preventing cellular senescence and apoptosis.

  • We added a new paragraph regarding study limitations in the Discussion section of the revised manuscript, addressing the limitations of our in vitro approach and the need for further in vivo studies (lines 494–506).
  • In the discussion, we expanded the discussion on the biological effects of changes in antioxidant markers (lines 494–506 in the revised manuscript).
  • This study was not subject to ethical review as it was an in vitro culture experiment using publicly available keratinocyte HaCaT cells.
  • The manuscript has been carefully reviewed by an experienced editor whose first language is English and who specializes in editing papers written by scientists whose native language is not English.

Reviewer 2 Report

Comments and Suggestions for Authors

The article is devoted to the study of bioactivity, in particular, antioxidant and protective action of the extract from Centella asiatica. The structure of the article is logically built and written in sufficient detail.

The first remark that is missing in this article is the characterization of the composition of the tested extract. It is necessary to provide a phytochemical analysis of the extract for the content of, for example, phenolics, flavonoids, alkaloids, tannins, steroids, and glycosides. Otherwise, it is completely unclear what substance was studied. This is the most critical shortcoming of this article.

The proposed design of the experiments and the parameters studied are quite sufficient for analyzing the protective effect of the extract on cell culture.

I have comments and suggestions:

1. This phrase in the abstract "This mechanism of action holds considerable promise for the prevention and improvement of skin aging" should be rewritten more carefully, given that the data on the protective effect were obtained only on keratinocytes, although there are other types of cells in the derma (fibroblasts for example).

2. I would like to ask to add to the manuscript microphotographs of keratinocytes in the control group, after exposure to peroxide and under the action of various concentrations of the extract.

3. There is no name of the line of human keratinocytes - was it HaCaT or some other?

In general, the article is well-formatted and logical.

At the same time, it is worth noting that I do not see any novelty in this work, given the lack of a complete characterization of what was studied. The biological activity of the extract has already been studied with confirmation of its composition (https://doi.org/10.1186/s40816-023-00353-8).

Author Response

Dear Reviewer 2:

We would like to extend our sincere thanks for your thorough review and positive feedback on our manuscript. Based on your comments, we have carefully revised the manuscript. All revisions are shown in red in the revised manuscript.

Below is a list of the revisions we have made based on your feedback.

POINT-BY-POINT RESPONSES TO THE REVIEWER 2 COMMENTS

Comment 1

The article is devoted to the study of bioactivity, in particular, antioxidant and protective action of the extract from Centella asiatica. The structure of the article is logically built and written in sufficient detail.

The first remark that is missing in this article is the characterization of the composition of the tested extract. It is necessary to provide a phytochemical analysis of the extract for the content of, for example, phenolics, flavonoids, alkaloids, tannins, steroids, and glycosides. Otherwise, it is completely unclear what substance was studied. This is the most critical shortcoming of this article.

Response to Comment 1

We provided the information on the ingredients of C. asiatica in the Materials & Methods section of the revised manuscript as follows (lines 100–105): Approximately, 124 chemical compounds including triterpenoids, polyphenolic compounds, and essential oils have been isolated and identified from C. asiatica [9]. Asiatic acid, madecassic acid, asiaticoside, and madecassoside are the major compounds identified in C. asiatica [10]. The extract contains 0.12% asiatic acid, 0.54% madecassic acid, 0.25% asiaticoside, and 1.02% madecassoside [5].

Comment 2

The proposed design of the experiments and the parameters studied are quite sufficient for analyzing the protective effect of the extract on cell culture. I have comments and suggestions:

  • This phrase in the abstract "This mechanism of action holds considerable promise for the prevention and improvement of skin aging" should be rewritten more carefully, given that the data on the protective effect were obtained only on keratinocytes, although there are other types of cells in the derma (fibroblasts for example).
  • I would like to ask to add to the manuscript microphotographs of keratinocytes in the control group, after exposure to peroxide and under the action of various concentrations of the extract.
  • There is no name of the line of human keratinocytes - was it HaCaT or some other?
  • In general, the article is well-formatted and logical. At the same time, it is worth noting that I do not see any novelty in this work, given the lack of a complete characterization of what was studied. The biological activity of the extract has already been studied with confirmation of its composition (https://doi.org/10.1186/s40816-023-00353-8).

Response to Comment 2

  • Thank you for your valuable opinion. To avoid definitive statements, we have revised it as follows: This mechanism of action may contribute to the prevention and improvement of skin aging (see lines 39–40).
  • In this study, we did not take microscopic photographs of keratinocytes. It is very difficult to conduct experiments that would allow us to provide photographs within the deadline for manuscript submission. In future studies, we would like to take morphological photographs. We would appreciate your understanding that we cannot respond to requests for comments.
  • The human keratinocyte cell line used in this study was HaCaT cells purchased from Cell Lines Service (DKFZ, Heidelberg, Germany). Thank you for pointing that out. That was our oversight. We have clarified it in "2.2. Materials for cell culture" in the revised manuscript (see lines 120–131).
  • The dysfunction and senescence of keratinocytes, which constitute most (>90%) of the epidermis, and the destruction of the skin barrier due to apoptosis can induce a decline in various skin characteristics, such as regeneration, moisturization, elasticity, and stretchability, and promote skin aging as evidenced by increased wrinkles and age spots, decreased moisturization, and decreased elasticity. Evidence suggests that 70% of these changes can be attributed to oxidative stress. Therefore, preventing and ameliorating the series of oxidative stress-induced apoptosis in keratinocytes could certainly prevent and ameliorate skin aging.

CA has an antioxidant effect, and its application to the treatment of various diseases related to oxidation, such as digestive disorders, wounds, nervous system disorders, and circulatory disorders, is being studied. As mentioned in the introduction section, research on the skin has been reported in relation to wrinkles around the eyes, atopic dermatitis, hair, collagen, and whitening of skin wounds.

There is already a lot of data on the effect of C. asiatica on the oxidation-antioxidation system. However, to the best of our knowledge, no study has yet examined in detail the effects of C. asiatica involving the series of reactions oxidative stress-induced cellular senescence and apoptosis in keratinocytes. Scientifically demonstrating the series of cell death defense mechanisms of C. asiatica on skin epidermal keratinocytes will be important evidence to open up future prospects for Centella asiatica extract as a preventive and improving agent for skin aging.

Therefore, we believe that our findings contain new evidence and will provide useful information to readers involved in skin aging.

Reviewer 3 Report

Comments and Suggestions for Authors

C. asiatica is used as a valuable traditional medicine in Southeast Asia and is becoming increasingly popular in the West. Because of its multiple health benefits (antimicrobial, antioxidant, anti-inflammatory, neuroprotective, and wound healing properties), C. asiatica is currently marketed as an oral supplement and as a topical ingredient in some cosmetic products. Several clinical studies have shown that topical application of C. asiatica can improve wound healing, as well as the course of atopic dermatitis and some other skin diseases. The inhibitory activity of C. asiatica and its fractions against pro-inflammatory and oxidative stress events may be the result of activation of antioxidant defense systems. The authors investigated the antioxidant and anti-apoptotic effects of C. asiatica in vitro using a keratinocyte cell line (HaCaT). The authors used standard methods to study the molecular factors of oxidative stress, antioxidant protection, and apoptosis. Despite the already available data on the effect of C. asiatica on the epidermis and on the oxidant-antioxidant system, the results obtained by the authors are new. These results may be useful for the readers of the journal Biology. Meanwhile, I have a few comments:

1) Section 2.7. Statistical Analysis. The statistical analysis methods used by the authors are only appropriate if there is a normal distribution. Please indicate how you determined the normality of the distribution of your data.

(2) Section 3 of the Results. Figure 2-6. The note should explain what the 2 and 3 * (asterisks) above the columns mean.

(3) Section 4. Discussion. It is advisable that the authors discuss in more detail the limitations of their study, including the presence of features of the HaCaT cell line, namely the presence of programmatic mutations in the genes of the transcription factor p53, involved in the regulation of apoptosis and oxidative stress and other cellular responses to DNA damage. About 124 chemical compounds, including triterpenoids, polyphenolic compounds and essential oils, have been isolated and identified from C. asiatica. However, the therapeutic role of individual components of C. asiatica is currently not fully understood. In addition, the authors ignored the possible skin side effects of using C. asiatica, although these effects are analyzed in available sources, including reviews: Torbati FA, Ramezani M, Dehghan R, Amiri MS, Moghadam AT, Shakour N, Elyasi S, Sahebkar A, Emami SA. Ethnobotany, Phytochemistry and Pharmacological Features of Centella asiatica: A Comprehensive Review. Adv Exp Med Biol. 2021;1308:451-499. doi: 10.1007/978-3-030-64872-5_25. PMID: 33861456; Bylka W, Znajdek-Awiżeń P, Studzińska-Sroka E, Dańczak-Pazdrowska A, Brzezińska M. Centella asiatica in dermatology: an overview. Phytother Res. 2014 Aug;28(8):1117-24. doi: 10.1002/ptr.5110. Epub 2014 Jan 7. PMID: 24399761.

Author Response

Dear Reviewer 3:

We would like to extend our sincere thanks for your thorough review and positive feedback on our manuscript. Based on your comments, we have carefully revised the manuscript. All revisions are shown in red in the revised manuscript.

Below is a list of the revisions we have made based on your feedback.

POINT-BY-POINT RESPONSES TO THE REVIEWER 3 COMMENTS

Comment 1

Section 2.7. Statistical Analysis. The statistical analysis methods used by the authors are only appropriate if there is a normal distribution. Please indicate how you determined the normality of the distribution of your data.

Response to Comment 1

The correction has been made as follows (Please see lines 320–324 in the revised manuscript): All data were presented as mean ± standard error of the mean (SEM). After confirming normal distribution of data, the significance of associations was determined by one-way analysis of variance and post-hoc analysis with Tukey’s multiple comparison test. Statistical significance was set at P < 0.05. These statistical analyses were performed using GraphPad Prism 9 (San Diego, CA, USA).

Comment 2

Section 3 of the Results. Figure 2-6. The note should explain what the 2 and 3 * (asterisks) above the columns mean.

Response to Comment 2

The meaning of the 2 and 3 * (asterisks) is explained in the footnotes of Figures 1–5 in the new version (which correspond to Figures 2–6 in the old version).

Comment 3

Section 4. Discussion.

(1) It is advisable that the authors discuss in more detail the limitations of their study, including the presence of features of the HaCaT cell line, namely the presence of programmatic mutations in the genes of the transcription factor p53, involved in the regulation of apoptosis and oxidative stress and other cellular responses to DNA damage.

(2) About 124 chemical compounds, including triterpenoids, polyphenolic compounds and essential oils, have been isolated and identified from C. asiatica. However, the therapeutic role of individual components of C. asiatica is currently not fully understood.

(3) In addition, the authors ignored the possible skin side effects of using C. asiatica, although these effects are analyzed in available sources, including reviews: Adv Exp Med Biol. 2021;1308:451-499. doi: 10.1007/978-3-030-64872-5_25. PMID: 33861456; Phytother Res. 2014 Aug;28(8):1117-24. doi: 10.1002/ptr.5110. Epub 2014 Jan 7. PMID: 24399761

Response to Comment 3

  • We added a new paragraph on limitations of this study to the Discussion section of the revised manuscript, in which we mentioned that elucidation of the apoptosis-triggering reactions, such as p53, remains as future study (see lines 499–506).
  • Thank you for your information. This information was provided as compositional information for asiatica extracts in the Materials and Methods section (see lines 507–512).
  • Thank you for the information about side effects. We provided the side effects information for asiatica in the discussion of the revised manuscript (see lines 486–493).

Round 2

Reviewer 2 Report

Comments and Suggestions for Authors

Thank you for your detailed answers to questions and comments.